# Plasma Oxidation Printing into DLC and Graphite for Surface Functionalization

**Tatsuhiko Aizawa [1],\***  **, Kenji Wasa [2] and Yoshiro Nogami [3]**

[1]  Surface Design Engineering Laboratory, SIT, 3-15-10 Minami-Rokugo, Ota-City, Tokyo 144-0045, Japan
[2]  MicroTeX Labs, llc., Ota, Tokyo 144-0045, Japan; wasa@mui.biglobe.ne.jp
[3]  Thermo-graphitics, Co. Ltd., Osaka 551-0031, Japan; yoshi_nogami_edosaki@yahoo.co.jp
\*  Correspondence: taizawa@sic.shibaura-it.ac.jp; Tel.: +81-03-6424-8615

**Abstract:** A diamond-like carbon (DLC) film, coated on a AISI420-J2 stainless steel substrate and vertically aligned graphite (VAG), was structured by high-density plasma oxidation to work as a DLC-punch for micro-stamping and DLC-nozzle array for micro-dispensing, in addition to acting as a copper-plated thermal spreader, respectively. Thick DLC films were micro-patterned by maskless lithography and directly plasma-etched to remove the unmasked regions. Thick VAG (Ca plates were micro-patterned by screen-printing and selectively etched to activate the surface. Raman spectroscopy as well as electric resistivity measurement proved that there was no degradation of VAG by this surface activation. Wet plating was utilized to prove that copper wettability was improved by this surface treatment.

**Keywords:** DLC-nozzle; DLC-punch array; vertically aligned graphite; surface-activated graphite; copper wet-plating

---

## 1. Introduction

Carbon has various solid morphologies in industrial materials—for example, graphene, diamond, diamond-like carbon (DLC), and carbon nano-tube CNT (Carbon Nano-Tube) in films; and graphite, diamond, and glassy carbon (GC) as a solid substrate. These carbon derivatives have high functionality in properties. For example, both graphene films and the vertically aligned graphite (VAG) comprising a stack of graphene planes have thermal conductivity higher than 1700 W/K/m [1,2]. However, they are too brittle to be mechanically machined for shaping into device and mechanical components. Even for joining these carbon films and solids with other materials, there are few processes other than chemical bonding [3] and composite structuring [4]. Surface functionalization via plasma technology and ultrashort pulse laser processing provide a means to shape these carbon films and solids into a mechanical part while keeping their original properties [5]. For example, a DLC film with a thickness of 5 μm was structured into a DLC-punch by a plasma oxidation process without loss of its smooth finish, chemical inertness, or high hardness [6]. GC was also machined into a mold with micro-textures for injection molding to transcribe the textures to optical plastics, using picosecond pulse laser micro-texturing [7].

In the present paper, this plasma oxidation oriented surface functionalization is explained and discussed by two applications. First, DLC films with a thickness of 10 μm were shaped into three-dimensional mechanical parts. Second, a VAG with a 1 mm thickness was surface-activated to improve its wettability in copper electroplating. The DLC-coated AISI420J2 substrate was prepared for shaping into DLC-punch and DLC-nozzle arrays by selectively etching the unmasked DLC. Scanning electron microscopy (SEM) was utilized to describe the shaped geometry with the measurement of the etching rate. A VAG specimen was micro-patterned by screen-printing and surface-activated to

physically modify its surface condition. Raman spectroscopy and electric resistivity measurement were employed to prove that no essential change took place, even on the activated surfaces. Electroplating was used to demonstrate that the copper wettability was improved in the wet plating to fabricate the copper-capped VAG heat spreader.

## 2. Experimental Procedure

### 2.1. Plasma Oxidation Etching System

A high-density plasma oxidation system was employed for micro-texturing of the carbon base materials. This system consisted of several parts (Figure 1), such as the vacuum chamber unit with the inner volume of φ55 cm × 80 cm, the panel control, the radio frequency (RF) and direct current (DC) power generators, the evacuation-pumping unit, and the mixture gas supply. Both the input and output powers were automatically matched by the frequency adjustment around 2 MHz since the chamber was neutral in electricity. Due to this automatic power matching, the oxygen plasma state was controlled in the response time of 0.1 ms. RF and DC plasmas were ignited independently to work in the experiment. The DC bias was controllable from 0 to −600 V, while RF voltage was controllable from 0 to 250 V. The pressure of oxygen gas was also tunable and controlled to be held constant with the tolerance of 0.1 Pa. As precisely explained in [8], a conductive stainless steel tube with the longitudinal cross section of 20 mm × 40 mm and the thickness of 2 mm was utilized as the hollow cathode to intensify the oxygen ion density up to the order of $10^{18}$ ions/m$^3$.

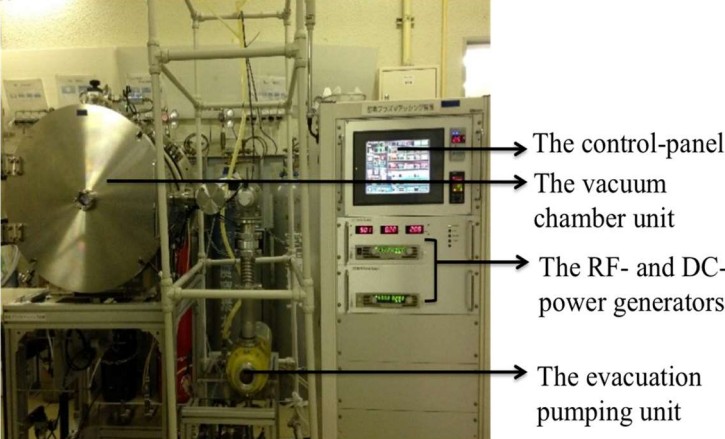

**Figure 1.** A radio frequency (RF) direct current (DC) plasma oxidation system for surface activation and micro-texturing experiments.

In the following experiments, the specimens were placed in the hollow cathode for selective plasma oxidation of the un-printed surfaces, as shown in Figure 2. Under this setup, the oxygen plasma sheath surrounded the specimen surface for efficient oxidation. The RF voltage was constant at 250 V, the DC-bias was −600 V, and the pressure was 30 Pa for diagnosis on the plasma oxidation. Oxygen gas purity was 99.9%. Since there was a risk of contamination by residual nitrogen gas, the experiments started after plasma diagnosis on the spectra of generated species in plasmas.

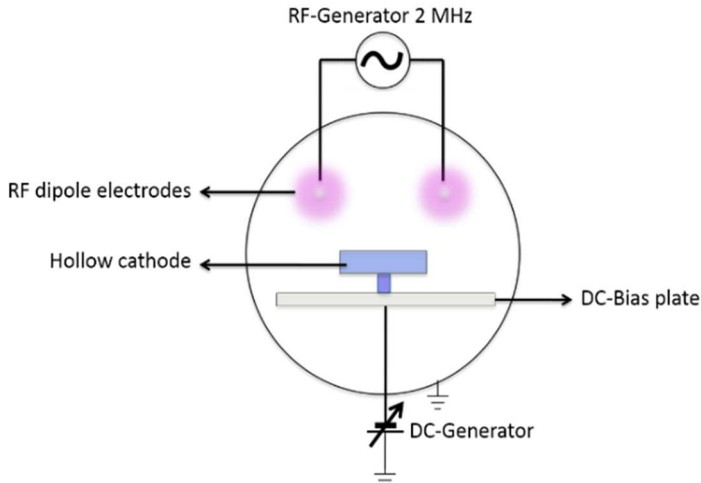

**Figure 2.** Experimental setup for plasma oxidation structuring.

*2.2. Preparation of Thick DLC-Coated Substrate and Thick Graphite*

Thick DLC films were coated on the AISI420J2 stainless steel substrates by MF-AC PECVD (Mid-range Frequency—Alternative Current Plasma Enhanced Chemical Vapor Deposition) process [9]. In the following experiments, the film thickness was constant at 10 μm, as depicted in Figure 3a. VAG bulk solid was also prepared by plasma-enhanced CVD as a c-axis-oriented pyrolytic graphite block with 300 mm × 300 mm × 20 mm [10]. The VAG specimen was cut and sliced from this block to have a size of 11 mm × 22 mm × 1 mm. Its surface was commonly roughed by using sand paper with the mesh of #1000; its average roughness was 1 μm. Figure 3b shows this VAG specimen after screen-printing. The details on this printing are defined later.

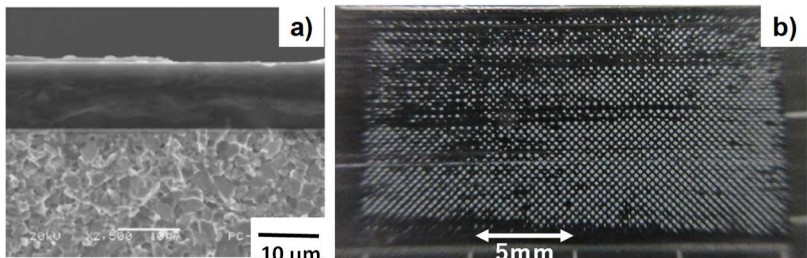

**Figure 3.** Carbon-based specimen. (**a**) Cross section of thick diamond-like carbon DLC-coated AISI420J2 specimen and (**b**) vertically aligned graphite (VAG) specimen with screen-printed micro-textures.

*2.3. Measurement and Observation*

SEM (SEM-1400, Nikon Co., Ltd., Tokyo, Japan) was used for observation of the processed specimens. Raman spectroscopy (InVia Qontor, Renishaw, Co. Ltd., Shinjuku, Tokyo, Japan) was employed for characterization of the local structure of VAG. Electric resistivity was also measured by the four-terminal method. The electroplating system (EPS-P2, Ebina-Denka, Co., Ltd., Ota, Tokyo, Japan) was utilized for the copper wet-plating test [11]. VAG samples were held on the electrode in the copper-base acid solution.

## 3. Results

*3.1. Plasma Oxidation Process*

Emissive-light optical spectroscopy (EOS; C8808, Hamamatsu, Co. Ltd., Hamamatsu, Shizuoka, Japan) was employed as an in situ plasma diagnosis method to describe the oxygen plasma state for plasma oxidation. A typical measured spectrum is shown in Figure 4 where the highest two

peaks are identified as activated oxygen atoms (O). Two strong significant peaks were detected at the wavelengths of 776 and 844 nm, which corresponded to the activated oxygen atoms. The peaks detected at 614 and 635 nm were also identified as oxygen atoms. This revealed that high oxygen flux could be utilized for the present oxidation process. This high yield of activated oxygen atoms in the present system is explained in the following. First, the oxygen molecules ($O_2$) were ionized by the electron detachment process: e.g., $O_2 \rightarrow O_2^+ + e$, where e denotes a free electron in the plasma. These generated electrons have more reaction cross section for other oxygen molecules by $O_2 + e \rightarrow O + O + e$. In addition, the generated $O_2^+$ was also ready to react by itself—e.g., $O_2^+ + e \rightarrow O + O$. Through these series of reactions, huge amounts of activated oxygen atoms were yielded by the present high-density plasma-induced chemical reactions. During the plasma oxidation etching, two carbon monoxide (CO) peaks were mainly in situ detected at the wavelengths of 212 and 256 nm and monitored. These peak intensities reduced monotonously with the etching duration time. CO peak intensity ratio might be dependent on the $sp^2/sp^3$ ratio in DLC and carbon bonding in VAG.

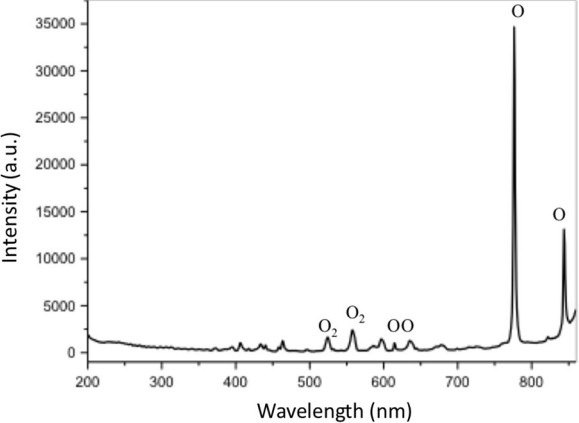

**Figure 4.** Emissive optical spectroscopy (EOS) for the plasma diagnosis of the oxygen species activated in the plasma sheath.

*3.2. Shaping to DLC-Punch Array and DLC-Nozzle Array*

Maskless lithography with the spatial resolution of 1 μm was utilized to print the two-dimensional micro-patterns. In a similar manner to [9], the nano-carbon and aluminum deposit prints with the total thickness of 140 nm were utilized for micro-patterning since they had sufficient erosion toughness not to be delaminated or broken away, even after plasma oxidation for 36 ks. A square dot pattern with the unit size of 3.5 μm × 3.5 μm was printed onto the DLC film surface with a pitch of 5 μm. Figure 5 depicts an SEM image of this micro-patterned DLC.

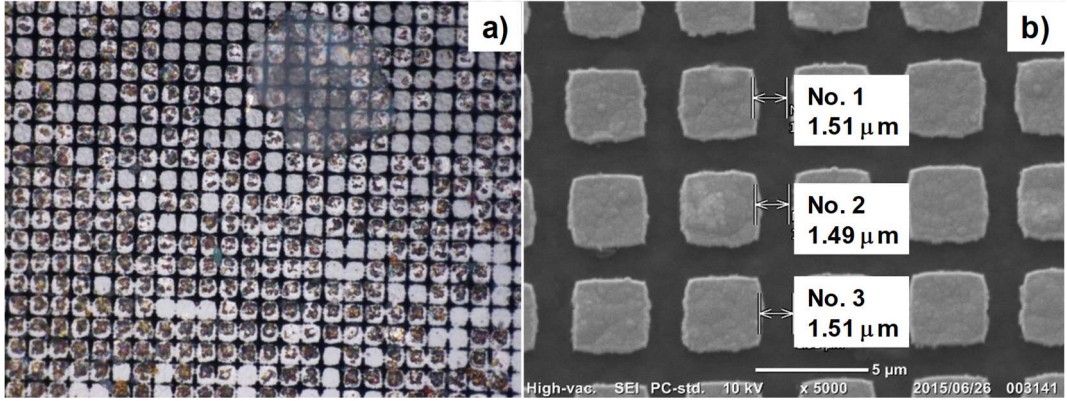

**Figure 5.** SEM image of micro-patterned DLC films. (**a**) A square dot alignment on the DLC film and (**b**) a square dot with the size of 3.5 μm × 3.5 μm.

These square dots were printed in regular alignment with the average pitch of 1.5 μm in correspondence to the CAD data for micro-printing by maskless lithography. Each square dot had a dimensional deviation in the sub-μm range. For example, the edge length expanded and shortened to 0.2 to 0.3 μm so that the clearance between adjacent dots ranged from 1.4 to 1.5 μm. This was because of the digital error coming from the limitation of spatial resolution in printing by the present maskless lithography. Figure 6a shows the DLC-punch array formed into the original DLC film by the plasma oxidation for 1800 s with the RF voltage of 250 V and the DC bias of −600 V. The average clearance of ten measured data in Figure 6 was 0.9 μm, shorter than the initial one in Figure 5. This shortage of clearance came from the burrs on the side surfaces of the DLC-punch. This shortage was further reduced to have the average clearance of 1.3 μm after plasma oxidation for 3600 s. These burrs and residuals were also removed by further oxygen flux from plasma sheath. Hence, besides these burrs, the original DLC film was sliced into a fine lattice structure with the line width of 1.5 μm and the pitch of 5 μm through plasma oxidation. As shown in Figure 6b, DLC-punches were arrayed with the uniform height of 3.1 μm.

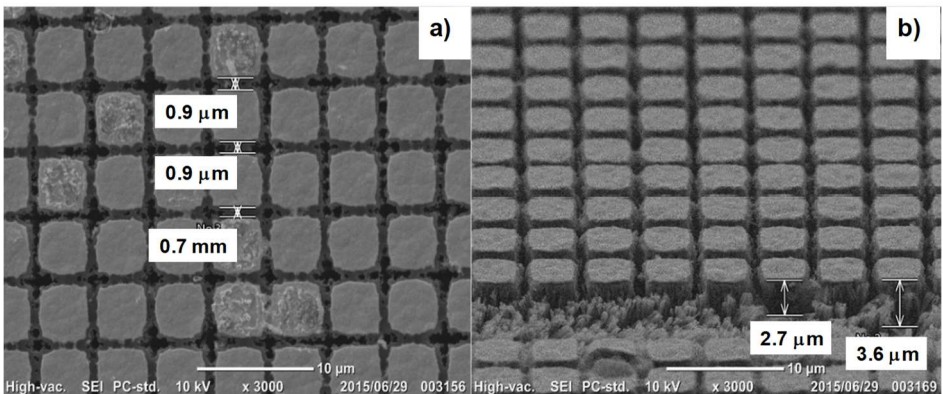

**Figure 6.** SEM image of DLC-punch array standing on the residual DLC film coated on the AISI420J2 stainless steel substrate. (**a**) Two-dimensional alignment of DLC-punches and (**b**) three-dimensional alignment of DLC-punches.

In a similar manner, the wider surface of the initial DLC film, except for the printed parts, was removed by the plasma oxidation to build up the DLC parts in correspondence to micro-printing. Maskless lithography was also employed to print the two-dimensional micro-patterns in Figure 7a (e.g., the circular dots with the diameter of 20 μm, having star-shaped, cross-shaped, and circular inlets, respectively). Figure 7b depicts the DLC-punch array after plasma oxidation at 60 Pa for 3600 s with the RF voltage of 250 V and the DC bias of −500 V. The unprinted DLC film with the thickness of 10 μm was almost removed to leave the DLC-nozzle array with three inlet geometries. DLC-nozzles had high hardness and heat resistance to work for dispensing.

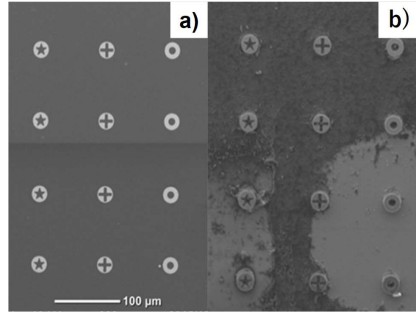

**Figure 7.** Fabrication of DLC-nozzle array from the micro-patterned DLC films. (**a**) Micro-patterned DLC film coated on the AISI420J2 substrate and (**b**) DLC-nozzle array standing on the substrate.

### 3.3. Selective Plasma Oxidation of VAG

The VAG specimen is a stack of graphene planes vertically aligned to form a top surface along the c-axis of the graphene plane. Its plasma oxidation behavior is different from amorphous carbon films without specific orientations. A simple masking technique was utilized to make selective etching of the VAG surface and to measure the etching rate. Figure 8 shows the SEM image and surface profile after plasma etching for 7200 s with the RF voltage of 250 V and the DC bias of −600 V. The unmasked VAG was etched away down to a depth of 26 μm. No etching took place on the masked region, while the unmasked VAG surfaces were homogeneously etched away. This revealed that the etching of VAG advanced anisotropically into the depth of the VAG to form micro-grooves in the VAG. The etching duration was varied to estimate the etching rate in this plasma oxidation conditions. Figure 9 shows the linear relationship between the etched depth of VAG and the duration; the etching rate of VAG was 13 μm/h.

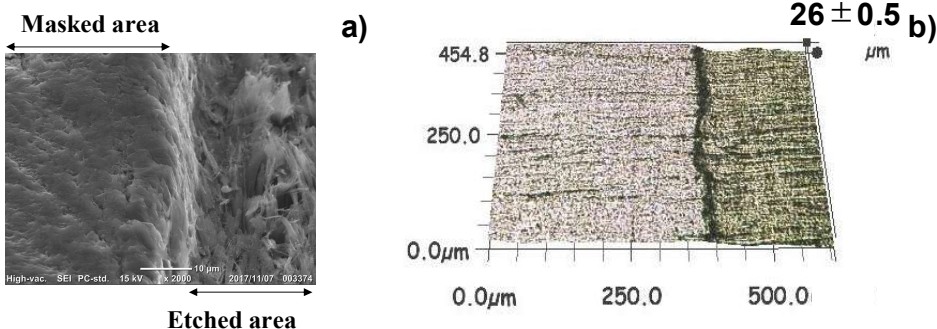

**Figure 8.** The etched surface of the VAG specimen. (**a**) SEM image (top view) and (**b**) three-dimensional depth profile.

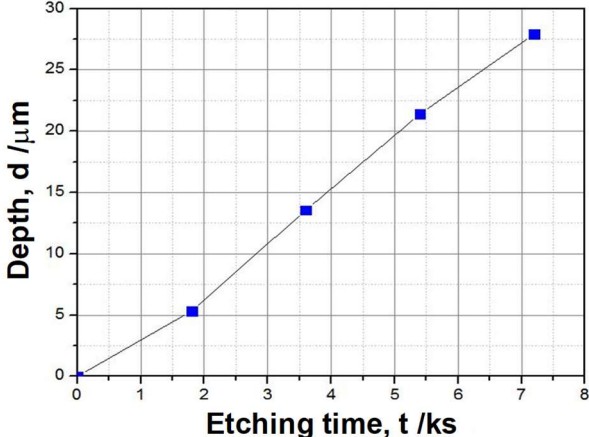

**Figure 9.** Relationship between the etched depth of VAG and the etching duration in plasma oxidation.

### 3.4. Surface Activation of VAG

The cross-line micro-pattern with each line width of 100 μm and skew angle of 45° was utilized to make geometric angulation on the VAG surface. Screen printing was utilized to print the skewed square dots as its negative pattern onto the VAG surface with use of $CaCO_3$ waiter solution ink. Figure 3b depicts the screen-printed VAG specimen. A unit cell of this printed micro-pattern was shaped by the regular diamond with the skew angle of 45° and the diagonal length of 150 μm. The width of lines between adjacent diamond patterns was 100 μm, in correspondence to the designed width of cross-lines.

After plasma oxidation for 1 ks and surface cleaning, electro-plating was utilized to investigate the wettability of copper film on the micro-textured and surface-activated VAG specimen surfaces. Without

additional treatments, the specimen was dipped into $CuSO_4$ solution and directly electro-plated. Figure 10 depicts the VAG surface condition after plating for 20 s. The copper precipitated layer was about to cover both surfaces besides the right-side edges, to be fixed at the electrodes. The maximum copper thickness was 1 μm. Copper deposited on the angulated surface, formed islands along the micro-grooves, and grew up to a uniform film surface. Figure 10 reveals this initial stage of copper nucleation and growth with the aid of the angulated surface and micro-grooves. This electro-plating process was continued up to 600 s to form the copper film on the whole VAG surfaces. The VAG specimen was clipped to the electrode for plating as depicted in Figure 11a. Figure 11b shows the copper-plated VAG specimen with the thickness of 30 μm. With the exception of the clipped surfaces, the whole VAG was completely plated with copper films. This demonstrated that the VAG surface was sufficiently physically modified to be fully copper-plated without additional chemical treatments.

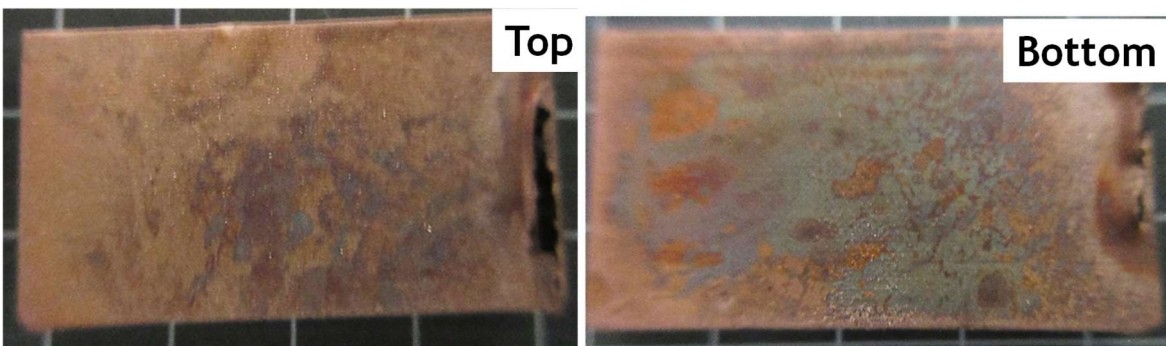

**Figure 10.** Surface of VAG specimen after electro-plating for 10 s. (**Top**) Copper deposition on the top surface and (**Bottom**) copper deposition on the bottom surface. The VAG specimen was fixed at the right-side edge to connect electrode by clipping.

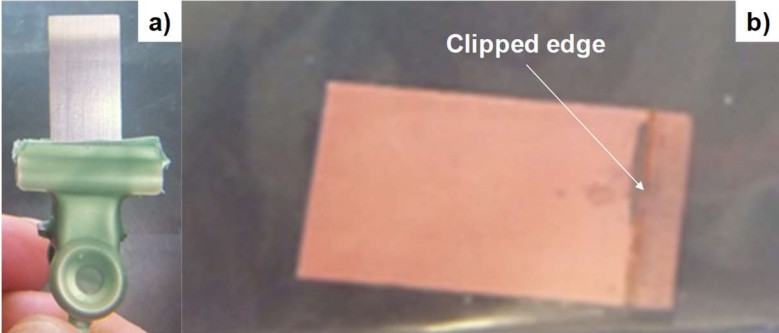

**Figure 11.** Electro-plating to cover the whole VAG specimens by the copper film. (**a**) Experimental setup and (**b**) plated VAG specimen for 600 s.

*3.5. Quality Check of Etched VAG*

The heat spreading capacity of the copper-plated VAG was dependent on the plasma-etched VAG property, as well as the thermal gap conductance between VAG and copper film. In the former, graphene plane alignment had the risk of deteriorating itself by plasma oxidation. Raman spectroscopy and electric resistivity measurement were employed to investigate the effects of oxygen etching on the VAG structure and property. Figure 12 depicts the Raman spectra for the original VAG surface and for the etched VAG surface. The typical graphitic peak—the so-called G-band—was detected with high intensity on both surfaces at 1580 and 1570 cm$^{-1}$, respectively. This proved that graphene planes with sp2-binding state were dominant even after plasma oxidation etching. Note that the D' band was also detected as a shoulder peak around 1620 cm$^{-1}$ besides the above G-band peak in the etched VAG surface. This might have been because of structural disorder in graphene planes as reported by [12].

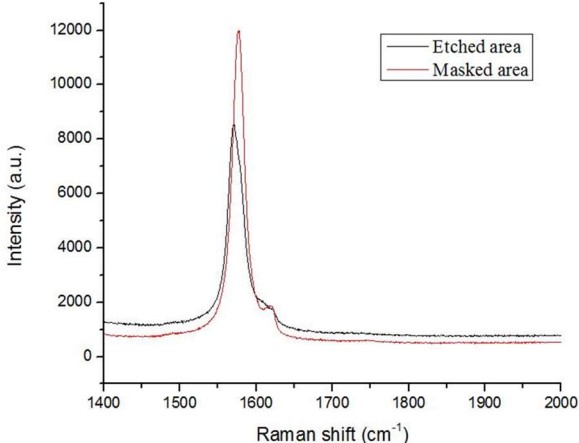

**Figure 12.** Comparison of Raman spectra between the masked and etched areas on the VAG surface.

The measured electric resistivity ($\rho$) of etched and masked areas were also compared with the reference data for graphene plane [5]—for example, $\rho$ = 1.17 $\mu\Omega$-cm for masked area, r = 1.25 $\mu\Omega$-cm for the etched area, and r = 0.7 $\mu\Omega$-cm in reference. The slight difference in the electric resistivity between the masked and etched VAGs proved that the etched VAG had the same graphene plane alignment as the original VAG, even after plasma oxidation etching. The higher electric resistivity measured on the etched VAG surfaces might originate from the disorder in the graphene planes in VAG. In the latter, the copper nucleation and growth into the micro-textured VAG-surface had a key to determine the thermal gap conductance. Under the mechanical joining of copper film to VAG, the thermal expansion mismatch between copper and VAG can be reduced by the local elongation of copper in the looser joints to VAG.

## 4. Discussion

The etching rate (*E*) of carbon derivatives by plasma oxidation is discussed with use of the reference data in [13]. PVD (Physical Vapor Deposition)-coated DLC and CVD-coated CNT films on Si wafers were prepared to measure the E for DLC and CNT films by using the same plasma oxidation system. *E* = 6 $\mu$m/h for DLC and *E* = 90 $\mu$m/h for CNT, respectively. In the present study, *E* = 6.2 $\mu$m/h to make DLC-punch from the thick DLC film CVD-coated on the AISI420J2, and *E* = 8 $\mu$m/h to make DLC-nozzle from the same DLC films. On the other hand, *E* = 13 $\mu$m/h when etching the VAG. Although the difference in the plasma oxidation conditions had a slight influence on this etching rate, E was mainly determined by the carbon structure. CNT film with sparse carbon structure was etched the fastest among various carbon derivatives. The VAG surface had sparse inter-graphene spaces, so it could be etched faster than amorphous carbon-structured DLC films.

Dimensional accuracy in shaping the DLC-punch and DLC-nozzle was mainly determined by the spatial resolution in micro-patterning onto the DLC films, in addition to the process controllability in plasma oxidation. In shaping the DLC-punch, the accuracy in its unit size and the clearance to other punches after plasma oxidation were within the spatial resolution of 1 $\mu$m in the maskless lithography to print the initial square dots. In forming the DLC-nozzle, its circularity and inlet geometries were reproduced by plasma oxidation in correspondence to the micro-patterned prints in Figure 5.

VAG was readily micro-textured and surface-activated with a faster etching rate than DLC films. Since *E* = 13 $\mu$m/h, the VAG surface was physically modified for 1 ks, enough to improve the copper wettability in electric plating. Little change took place in microstructure and electric properties even after this modification. This copper-capped VAG had sufficiently low gap in thermal conductance between VAG and copper film as a heat spreader.

## 5. Conclusions

Plasma oxidation provided a tool for the machining and shaping of carbon-base materials. A DLC-punch array with the unit size of 3.5 μm × 3.5 μm × 3.1 μm was formed by fine machining of thick DLC films with the slicing width of 1.5 μm. The DLC-nozzle array with the outer diameter of 20 μm was shaped by removing the unprinted DLC surface. Various inlet geometries were accommodated to this DLC-punch array in order to dispense high-viscosity polymers and lead-free solders.

Vertically aligned graphite was also micro-textured and surface-activated to have suitable angulation to improve copper wettability in electric plating. The whole VAG surface was copper-plated with the thickness of 30 μm. Since no significant change was detected in microstructure and electric properties after this processing, this copper-plated VAG is expected to work as a heat spreader in the packaged system.

**Author Contributions:** T.A. developed the research plan together with K.W., made plasma oxidation structuring experiments with Y.N., and wrote this research article.

**Funding:** This study was financially supported in part by the METI-program on the supporting industries in 2018.

**Acknowledgments:** The authors would like to express their gratitude to H. Tamagaki (Kobelco, Co. Ltd.), H. Nakata (Ebina Denka Kougyo, Co. Ltd.), Imron Rosadi (University of Brawijaya, Indonesia) and N. Watanabe (Shibaura Institute of Technology) for their help in the experiments.

**Conflicts of Interest:** The authors declare no conflict of interest.

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
