# Peer review of "Plasma Oxidation Printing into DLC and Graphite for Surface Functionalization"

_carbon, 2018_

Round 1

Reviewer 1 Report

The present manuscript “Plasma Oxidation Printing into DLC and Graphite for Surface Functionalization” describes a surface functionalization via a high density plasma oxidation process for two different substrates. Firstly a DLC coating previously deposited on a stainless steel to achieve micro-structures for e.g. stamping etc. and secondly a vertically aligned graphite (VAG) plated with copper after the plasma process for better heat conductivity.

The organization of the manuscript itself is reasonable and follows the rules of the journal. But, there are some points, which should be addressed in a revised manuscript before publication. Therefore, I recommend major revision of the manuscript based on the following comments, as it is currently not publishable:

Introduction:

-          The thickness of the films and VAG plates should be included for clearness, both is only stated as “thick” (? nm or µm or ?… range)

Keywords:

-          The given keywords repeat in parts the title, e.g. DLC, plasma oxidation, surface activation (=functionalization). Be more specific here and restrict to those that achieve added value.

Introduction:

-          The introduction is very condensed, which is not bad, but six out of ten references are already self-citations. Excessive self-citation should be avoided, be more objective here. Plasma oxidation processes are a common way of surface functionalization / activation for further materials improvement.

-          Furthermore eight out of 14 references are self-citations: nearly sixty percent!!! Be more specific in which of your own references are really needed.

Experimental Procedure:         

- Line 51-54 text redundancy (can be removed or only mentioned here …)

Compare Line 41-43: High density plasma nitriding system is first stated with comments on the hollow cathode device to intensity the oxygen ion density.

Not needed e.g. in the introduction (as it is just a statement) or rearrange these sentences and use it better only in the experimental part …

-          Description of the etching system: Here I miss details on the dimensions of the chamber, parameters of the experiment, e.g. working pressure of chamber (source of contaminations e.g. N2), evacuation resp. the starting pressure to avoid contaminations, purity of the oxygen gas, mass flow controller for constant pressure, distance of electrodes to the bias plate and resp. the hollow cathode, use of inert gas etc.

Be here more specific and deliver more details on parameters not on parts of the system, which are common sense (pumping unit, control panels!!!) as this description does not deliver any details of themselves either …

-          Furthermore, as stated in the introduction “comments on the hollow cathode”, I really miss here details on this specific device. There are no dimensions or shape (hollow…) of the device given, nor any (brief) advantages/disadvantages of this set-up compared to others.

-          Fig.1: The caption indicates: A plasma oxidation system …

Is this the chamber of this experiment or any exemplary one? Be more specific!

-          What is the dimension or max. possible one of the specimens resp. substrates.

-          The printing process is not described nor the appearance before plasma oxidation, especially for the specimen Fig. 3b + morphology after grinding, Furthermore, Fig. 3 needs more description: a) is a cross section and b) a top view of two different substrates. Why not consistent top and cross views for both?

-          How the film thickness was determined, the preparation, device description, error tolerance etc. …

-          In the experimental procedure no machine type is given for the SEM, Raman (only for Cu-coatings), EOS (even not mentioned but given by Fig. 4), thickness determination and resistivity check, nor any details on these measurements or sample preparations etc.!!!

-          Is there any attempt done to check the types of generated O-bonding by FT-IR or XPS. In EOS only O and O2 are given but no allocations to C-species. It would be also interesting how the sp2 to sp3 ratio and C-bonding (C-O) change by the etching process in both samples. In this context it would also be interesting to know the penetration depth of the Raman system?

-          No details are given for the copper-plating or reference for the copper-base acid solution in Line 124

-          First paragraph after 3. Experimental results is again somehow redundant and a repetition in itself (Line 127-129).

-          Whole manuscript: Please use seconds (s) and hours (h) as it is more common than ks

-          In Fig. 5 b, 6 a and b, the numbers are too small to see, please enlarge the font for distances and heights etc. and deliver error margins for the distances and heights in the text

-          Fig. 7: How can one see the removement of 10 µm DLC via a top view, a cross view o an angle view would be more helpful? How the SEM samples were coated, C or Au, and thickness?

-           

-          Fig 8b: remove the 606.3 µm, plus deliver the error tolerance for the height of 26.22 µm. Furthermore, I think one digit after the comma should be enough, also for the EOS peak nomination; check the entire document.

-          Line 242-243: Both surfaces are uniformly …

A uniform coating cannot be concluded from Figure 10 Bottom …

-          Line 296: …the same physical properties as…

Be more specific here. It is only proven for the resistivity, but not for the wettability or morphology etc.

-          Line 304-305: The etching rate (E) of carbon derivatives by the plasma oxidation is discussed with use of the reference data in [14].

I do not understand this sentence here as no additional statement here is needed if it is discussed on the basis of [14]??? Furthermore the substrates are different and please deliver the information, if the deposition is the same as it is stated the DLC was there PVD-coated and the CNTs CVD. The present sample is RF-CVD coated. More information is needed to ensure comparability.

-          I do not get the point of the achievement of the high thermal conductivity here, be more specific in this or deliver more information or comparison. It is not clear to me, as it is stated in Line 323.

-          Line 330: … in order to dispense the viscous agents and metallic melts.

Is this one possible application for these nozzles, as it was not mentioned before? At the moment it is without context.

Some remarks on sentences with linguistic suggestions or where rework is needed to clarify the statement:

-          Line 15: … to make surface activation.

Better: … to activate the surface.

-          Line 31: … these carbon films and solid …

I suppose it should be the plural: …these carbon films and solids …

-          Line 40: … discussed through two applications …

Better: … discussed by two applications …

-          Line 40: … two applications; e.g. …

As the applications are specified I would recommend to use a) … and b) not e.g.

-          Line: 44: … SEM …

The abbreviation is explained later in the text, should be done here at the first appearance

-          Line 57: This system consisted of …

Better: This system consists of …

-          Line 62: … plasmas were ignited independently to be working in experiment.

Better: … plasmas were ignited independently to work in the experiment.

-          Line 90: … MF-CVD process

Not RF-CVD?

-          Line 93: …20t mm3 and …1t mm3

What does the t mean?

-          Line 123-124: The electroplating system was utilized for copper wet-plating test.

Redundancy, already mentioned …

-          Line 135-136: The other peaks …

Rearrange somehow, it sounds strange … like: Which ones?

-          Line 138: … plasmas.

I think it should be singular here: … plasma.

-          Line 179: As shown in Fig. 5 b), …

I think it should be Fig. 6b)

-          Line 194-195: … 60 Pa … DC-bias of -500 V.

Really 60 Pa? In the beginning it was stated the pressure is constant 30 Pa, and -500 V, as the rest is consistently -600 V.

-          Line 232+236: superscript for angle dimension: … 45°

Author Response

The whole texts and Figures were revised to make the paper more comprehensive.  In particular, Introduction was revised to explain the aim and background of this study.

Reviewer 2 Report

The paper "Plasma Oxidation Printing into DLC and Graphite for Surface Functionalization" discuses the use of low pressure RF plasma for structuring diamond like carbon (DLC). In addition, plasma treatment was also used for surface activation of vertically aligned graphite (VAG) to improve wettability of the surface for copper plating. It is not quite clear in the paper what the end application for these surfaces are.

This paper is poorly written, not very well presented and not very well structured. The paper is too brief; in the introduction more background work from literature need to be given and the aims of the paper need to be clearly mentioned. In the experimental section, enough details need to be given for the plasma setup and all equipment used for data collection and analyses. Experimental parameters applied to perform experiments and collect data need to clearly mentioned in their relevant places. The results are briefly discussed therefore adequate interpretation is needed for that many Figures in the Results and Discussion section.  The abstract and conclusion sections need to be expand a little and capture the results presented in the paper.

Also, the paper need a good proofreading as there are many typographical errors and mistakes in grammar, style, and spelling. Abbreviation should not be used in the title.The authors used "ks" as a unit for the time i think, a "min" should be used instead.

Author Response

The texts related to experiments and conclusion were revised with revision of Figures.  References were also updated to reduce the number of self-citing papers and to cite other papers.

Round 2

Reviewer 1 Report

The present manuscript “Plasma Oxidation Printing into DLC and Graphite for Surface Functionalization” improved according to the implementation of reviewer comments.

Unfortunately, there are still some points the authors should address before accepting the manuscript for publication.

Keywords:

-          The given keywords still repeat in parts the title, and some are deleted which could be of an added value:

Repetition of: Plasma oxidation,(can be deleted)

Is deleted, but has an added value: vertically aligned graphite (VAG)

 Experimental Procedure:

 -          Description of the current and used etching system improved, the only thing I miss is the purity of the oxygen gas (could be a source of contaminations e.g. N2…).

-          In the experimental procedure only the brand names of the supplier for the machines are given but not its specific type/number, nor any details on these measurements (like kV or distance used in the SEM) or sample preparations etc.!!!

By the way it is “Renishaw” for Raman not "Renishow"

-          For the question: Is there any attempt done to check the types of generated O-bonding by FT-IR or XPS. In EOS only O and O2 are given but no allocations to C-species. It would be also interesting how the sp2 to sp3 ratio and C-bonding (C-O) change by the etching process in both samples. In this context it would also be interesting to know the penetration depth of the Raman system?

Your answer is:

Figure 4 describes the oxygen nuclei generated in the plasma sheath.

 That’s correct, but it does not answer the question according to the types of bonding in respect to give details or the sp2 to sp3 ratio, nor any information to the penetration depth of the Raman system!!! Please be more specific to the question, maybe it was not in the focus of this work.

Some remarks on sentences with linguistic suggestions or where rework is needed to clarify the statement:

-          Line 31: … these carbon films and solid …

I suppose it should be the plural: …these carbon films and solids …

-          Line 93: still …20t mm and …1t mm

What does the t mean? Is it for thick? I think it is not necessary:

-          Reference 8 needs revision: it is not issue 46, but 86-87, (Part2) … please revise.

Author Response

Minor revision was made with answer to question.

Reviewer 2 Report

The authors have made sufficient changes to improve the quality of the manuscript. Therefore, I recommend it for publication in C.   

Author Response

Minor revision was made in the text including the answer to the question.